# The Adverse Effects of TiO_2_ Photocatalycity on Paraloid B72 Hybrid Stone Relics Protective Coating Aging Behaviors under UV Irradiation

**DOI:** 10.3390/polym13020262

**Published:** 2021-01-14

**Authors:** Wenjuan Li, Junling Lin, Yaru Zhao, Zihe Pan

**Affiliations:** 1College of Art, Taiyuan University of Technology, 209 University Avenue, Jinzhong 030600, China; junling_lin98@163.com (J.L.); zhaoyaru123rhea@163.com (Y.Z.); 2Institute of Resources and Environmental Engineering, Shanxi University, 92 Wucheng Road, Taiyuan 030006, China

**Keywords:** hybrid protect coating, TiO_2_, aging

## Abstract

The incorporation of photocatalytic nanomaterials into polymer coatings is used to protect stone relics from weathering. However, the photocatalytic nanomaterials might generate excess free radicals to degrade the polymer matrix. In this work, a certain amount of TiO_2_ nanoparticles were dispersed into Paraloid B72 and applied onto sandstone relics to explore the adverse effects of TiO_2_ nanoparticles on Paraloid B72 under ultraviolet (UV) irradiation. To fulfill this goal, the effects of TiO_2_ on pore formation and the structure of Paraloid B72 was studied by scanning electron microscopy (SEM). Moreover, the surface chemical composition, pore structure, surface roughness and surface wettability were explored via Fourier transform infrared (FTIR) spectroscopy, SEM, optical profilometer and water contact angle measurement under UV irradiation. Results showed that the incorporation of TiO_2_ nanoparticles prohibited the generation of pores in Paraloid B72 and there were no pores formed when the content of TiO_2_ exceeded 0.8 wt%. The water contact angle of origin Paraloid B72 and TiO_2_/Paraloid B72 decreased with the prolonging UV irradiation. Moreover, TiO_2_ nanoparticles were extracted from the matrix and the pores cannot be detected with the prolonging UV irradiation time under a higher content of TiO_2_. These research findings might promote the understanding of using photocatalytic nanomaterials in developing stone relics’ protective coating.

## 1. Introduction

Stone relics are an important type of heritage which record the life of ancient people and offer a better way for later generations to learn the development of human society [1,2,3]. Thus, the importance of heritage stones protection has attracted extensive attention worldwide. China has a long history over 2500 years and has a great many stone historical relics across the whole country [4,5,6,7,8,9]. Shanxi province located in the northwest part of China is regarded as the cradle of Chinese civilization and is famous for its cultural relics, especially the stone relics e.g., Yungang Grottoes [10]. However, these stone heritage relics currently experience severe weathering and aging due to increasing air and water pollution (SO_2_, SO_3_, NO_x_ etc.) in Shanxi over the past decades. Especially, the organic compounds which are emitted from the coal burning and utilization leading to more apparent damage to the stone relics [11,12,13,14,15]. The protection of the stone heritage relics becomes increasingly urgent.

Although a great many techniques have been developed to protect the stone relics from environment weathering [16,17,18], many challenges still need to be solved. Coating, especially polymer hybrid coating with superhydrophobicity and self-cleaning property is regarded as one of the most effective approaches to improve anti-pollution and reduce water penetration performances by [19,20,21,22,23]. Ferri et al. [24] added different amounts of SiO_2_ nanoparticles into polysiloxane to increase the surface roughness and hydrophobicity which was applied on limestone, sandstone and granite reaching durable water repellency, thereby protecting the stone relics. However, the polymer hybrid composite coatings are easily aged in the environment leading to yellowing, cracking, and peeling, even blocking the micro-pores [25,26]. Another method to protect the surface from organic pollutants is adding photocatalytic nanomaterials into polymer coatings to degrade the organic compounds which improves the anti-aging performances of protective coatings [27]. Ruffolo et al. [28] incorporated photocatalytic ZnO and ZnTiO_3_ nano powders into Paraloid B72 to degrade the pollutants and microbial. Fernández et al. [29] reported using photocatalytic ZnO/MgO- potato dextrose agar composite coting to protect the stone heritages from fungi calcareous by UV photo-degradation. TiO_2_ has been widely reported to be used in stone heritage relic protection coating due to its excellent photoactivity under ultraviolet (UV) light [30,31,32]. Russa et al. [33] reported that the anti-aging and self-cleaning performances of acrylic, Paraloid B72 and Akeograd P composite coatings were improved by adding a certain amount of TiO_2_ and the tested methylene blue was degraded within 5 days under UV light. Quagliarini et al. [34] indicated that only adding TiO_2_ into polymer coating is not effective to improve the self-cleaning and anti-fouling performances. This is because of a low amount of TiO_2_ showing less photo degradation efficiency while a large loading of TiO_2_ caused the opaque of the coating.

Although TiO_2_ can be easily activated under UV light generating free radicals such as, O_2_^•^ and OH^•^, these free radicals are high in energy which can destroy the chemical bonds of polymers thereby degrade organic pollutants [35,36]. Nevertheless, the high-energy free radicals are more likely to break the chemical bond of polymers and form new bonds leading to the degradation of the TiO_2_-polymer hybrid coatings [37,38,39]. The degradation routes are proposed as follows:(1)$$R\cdot +O_{2}\overset{hv}{\to }RO_{2}\cdot$$
(2)$$RH+O_{2}\overset{hv}{\to }R\cdot +HO_{2}\cdot$$
(3)$$ROOH\overset{hv}{\to }RO\cdot +\cdot OH$$
(4)$$R\cdot +O_{2}\to RO_{2}\cdot$$
(5)$$RO_{2}\cdot +RH\to ROOH+R\cdot$$
(6)RO·+RH→ROH+R·
(7)R·+R·→R−R
(8)$$R\cdot +RO_{2}\cdot \to ROOR$$
(9)$$RO_{2}\cdot +RO_{2}\cdot \to ROOR+O_{2}$$

Auffan et al. [40] reported that Al(OH)_3_ and polydimethylsiloxane (PDMS) were peeled off from the surface of TiO_2_ after exposure to UV light in water and the surface of PDMS was oxide indicating the generated radical oxygen species degrade the protecting polymers. Santaella et al. [41] coated TiO_2_ with Al(OH)_3_ and silicon-based polymer and tried to preclude the generation of radical oxygen species and the decomposition of the polymer. Therefore, it is critical to study the adverse effects of the photoactivity of TiO_2_ in cultural heritage protective polymer coatings.

In this work, we hypothesized that the incorporation of high photoactive TiO_2_ nanoparticles might cause the degradation of a polymer matrix leading to photo aging. To verify this, varied amount of TiO_2_ nanoparticles were dispersed into Paraloid B72 solution homogeneously and coated onto sandstones. The degradation behaviors of TiO_2_/Paraloid B72 hybrid coating under UV light irradiation was studied. Experimental results demonstrated that increasing the loading of TiO_2_ nanoparticles leads to the difficulty of pore generation in Paraloid B72 and the incorporated TiO_2_ nanoparticles might block the pores. Moreover, the generated nano-sized pores were vanished with prolonging the UV irradiation time. This phenomenon was verified from the surface wettability and roughness showing that with the expanding of UV irradiation the surface became more hydrophilic and more peaks and valleys were formed. The research findings might promote the fundamental understanding of anti-aging, self-cleaning and photodegradation of a polymer matrix of photocatalytic TiO_2_ nanoparticles in stone cultural heritage polymeric coatings.

## 2. Experimental

### 2.1. Materials

Commercial P25 (Shanghai Maclin Biochemical Co. Ltd., TiO_2_ nanoparticles, 20 nm), Paraloid B72 (Dow Chemical Company, Midland, MI, USA), deionized water (DI water, supplied by the lab), sandstone (Taiyuan Art Stone Company, Taiyuan, Shanxi, China), acetone (AR, Shanghai Maclin Biochemical Co. Ltd., Shanghai, China), tricloromethane (AR, Shanghai Maclin Biochemical Co. Ltd.), were used as received without any further treatment.

### 2.2. Preparation of TiO_2_-Paraloid B72 Composite Coating

Two types of coating were prepared for comparison: pure Paraloid B72 coating and Paraloid B72 incorporating different amounts of P25 coating. The preparation of Paraloid B72 coating followed the literature [31] that 1.2 g Paraloid B72 was dissolved into 50 mL tricloromethane under vigorous stirring until Paraloid B72 was dissolved completely. P25 was first dispersed into Paraloid B72 solution and ultra-sonicated for 30 min. Then, the sandstone sample was immersed into the above solution for coating and was taken out after 5 s and dried in an ambient environment at room temperature. Three to five samples were prepared for each group specimen for repeating to obtain relatively accurate results. The prepared samples were named as follows: P25 dispersed into Paraloid B72 was named TiO_2_/B72 (Table 1).

### 2.3. Characterization Methods

The prepared specimens were aged under UV light (wavelength of 254 nm, WFH-203B, Hangzhou Qiwei Equipment ltd. Com., Hangzhou, China) for 72 h, 120 h and 240 h, respectively, to characterize the effects of TiO_2_ nanoparticles on photo aging behaviors of Paraloid B72 composite coatings. For each ratio group, at least three samples were tested to obtain solid data. The power of the UV light we used in this work was 6 W. The thermic effect of this UV light was negligible which did not generate significant amount of heat leading to the increase of temperature on the samples. The radiation intensity of the UV lights was measured via radiometer (FieldMaxⅡ-TO, COHERENT, Portland, OR, USA) and approximately 140 data were collected and the average radiation intensity was calculated as 556.3 µW/cm^2^.

The dispersity of TiO_2_ in Paraloid B72 composite coatings and the stone were characterized via scanning electron microscopy (SEM, JEOL JSM-7001F, Tokyo, Japan). Since the stone was hard to cut into small and flat pieces, we applied the coating onto a flat matrix (e.g., glass slide, Cu plate, silicon wafer) to carry out the surface morphology characterization before and after UV irradiation. The chemical composition of the stone used in this work was measured by X-ray fluorescence spectrometer (XRF, PW4400, Karlsruhe, Bruker). In order to compare the changes of the coatings before and after the photo aging, a Fourier transform infrared spectrometer (ATR-FTIR, a PerkinElmer 2000 spectrometer, Waltham, MA, USA) was used to characterize the surface chemistry of the original and photo aged samples. Water adsorption of pure sandstone and Paraloid B72 coated stones, TiO_2_/B72 coated stones after UV irradiation was calculated as follow [42,43]:(10)$$g=\frac{G}{A}=\frac{∆m}{∆t\times A}$$

In which, Δ*m* is the weight change before and after water adsorption for certain time (mg); Δ*t* is the adsorption time (h); A is the surface area of the tested specimens (m^2^).

The surface wettability of Paraloid B72 coated stones and TiO_2_/B72 coated stones before and after UV irradiation and under varied UV irradiation time was measured in terms of water contact angle to illustrate the effects of TiO_2_ on surface wettability. Typically, a drop of water (10 μL) was placed onto the sample and the water contact angle was recorded via water contact angle measurement instrument (Krüss DSA25, Hamburg, Germany).

## 3. Results and Discussion

### 3.1. Chemical Composition and Surface Morphology of Sandstone

The chemical composition of the stone used in this study was characterized through X-ray fluorescence (XRF) (Table 2) showing that the stone was composed of SiO_2_, Al_2_O_3_, Na_2_O, CaO, K_2_O and Fe_2_O_3_ et al. The main composition of the stone is SiO_2_ with the composition of 84.20 wt% while only 1.78 wt% of Al_2_O_3_ was detected in the specimen. Though other components were found, there composition was very small especially the static errors were much higher than their composition indicating the incorrect value. Thus, the main composition of the stone was SiO_2_.

The sandstone was cut into 3 cm × 3 cm × 2 cm (length × width × height) (Figure 1a) and the electronic image of the sandstone pieces showed that the surface was not smooth. The surface morphology of the stone was characterized by SEM (Figure 1b–d) showing that the sandstone was not smooth (Figure 1b) and it was composed of many flakes with the length of around 1 to 3 μm (Figure 1c) while the thickness was less than 100 nm (Figure 1d). These accumulated nano flakes formed a compact laminar microstructure and microporous structure. This unique structure enabled the mechanical strength and the gas and water permeation property which is a promising option for stone heritage material.

### 3.2. Scanning Electron Microscope (SEM) Characterization of TiO_2_-Paraloid B72 Coated Sandstone before Ultraviolet (UV) Irradiation

Surface morphology of origin Paraloid B72 and the effects of TiO_2_ on Paraloid B72 coatings are shown in Figure 2. There are many pores ranging from 100 nm to around 500 nm on origin Paraloid B72 (Figure 2a,b) enabling the well gas and water permeability. The formed pores are attributed to the phase invasion effects of Paraloid B72 after the evaporation of the solvent (e.g., acetone, trichloromethane). The pore structure was well kept after adding 0.2 wt% and 0.5 wt% TiO_2_ nanoparticles into Paraloid B72 and the pore size is similar as the origin Paraloid B72 indicating that the negligible effects of TiO_2_ nanoparticles on porous structure at low content (Figure 2c,d). Nevertheless, adding more TiO_2_ nanoparticles into Paraloid B72 caused significant porous structure change, e.g., adding 0.8 wt% TiO_2_ blocked the pores and closed-pores were formed (Figure 2e). Continuously increasing the loading of TiO_2_ nanoparticles into Paraloid B72, no pores were observed after adding 1.0 wt% TiO_2_ nanoparticles and significant TiO_2_ nanoparticles aggregations were observed on Paraloid B72 (Figure 2f).

### 3.3. Surface Wettability of UV-Aged TiO_2_/B72 Hybrid Composite Coatings

TiO_2_ was widely used to degrade the organic pollutions which were adsorbed or deposited on the protective coatings via its photocatalytic property to maintain the surface cleaning. However, TiO_2_ can not only degrade the surface organic pollution compounds but also it can photo-degrade the polymer coating matrix because of its strong oxidation property. Herein, the surface wettability of TiO_2_ reinforced Paraloid B72 coatings was measured and compared with the pure Paraloid B72 coating. The wettability in terms of water contact angle before and after the UV photo irradiation was measured to illustrate the function of TiO_2_ nanoparticles. As shown in Figure 3a, water contact angle of the as-prepared specimens is larger than 137° and the homogeneously dispersed TiO_2_ decreased the water contact angle to higher than 106° due to the hydrophilicity of TiO_2_ nanoparticles. Nevertheless, the water contact angle decreased with the prolonged UV treatment indicating that UV irradiation caused surface changes leading to the decrease of the water repellent property. The water contact angle dropped to lower than 60° after 7 days UV irradiation (Figure 3a,b) showing the notable adverse effects of TiO_2_ on Paraloid B72. A huge reduction (40% and 54%) was observed on Paraloid B72 and TiO_2_ composite coatings. By comparison, the water contact angle of Paraloid B72 dropped from 137° to around 92° after exposed to UV light for 7 days indicating the degradation of Paraloid B72 under UV light (Figure 3a). These results showed that incorporating TiO_2_ into Paraloid B72 might generate adverse effects on Paraloid B72 under UV irradiation. The water contact angle decreased with the increase of aging time for all samples and this phenomenon became more apparent with the increment of the loading of TiO_2_ nanoparticles in Paraloid B72. For the composite coatings with TiO_2_ content of 0.8 wt% and 1.0 wt%, the surface wettability decreased significantly with the prolonging aging time under UV light. To investigate the mechanism behind the phenomena, FTIR, SEM, optical profilometer and water adsorption before and after UV irradiation were performed in the following sections.

### 3.4. Surface Chemistry of UV-Aged TiO_2_/B72 Hybrid Composite Coatings

FTIR is a common tool in characterizing the surface chemical composition. In this study, FTIR spectrum was recorded before and after UV irradiation (Figure 4). As shown in Figure 4a, the characteristic peaks of Paraloid B72 and TiO_2_/B72 composite coating were similar before UV irradiation. CH_3_, C-C skeleton vibration and CH_3_ rocking vibration were found in these specimens at the wavelength number of 758, 836, 858 and 968 cm^−1^, respectively (Figure 4a and Table 3). A weak peak located at 1020 cm^−1^ was the rocking vibration of C–H which was only observed in 0.8TiO_2_/B72 and 1.0TiO_2_/B72, respectively. C–O–C stretching vibration and C–C stretching vibration locate at the wavelength of 1020 cm^−1^ and 1160 cm^−1^ were only detected in TiO_2_/B72 hybrid coatings (Figure 4a and Table 3). C–O–C vibration and δ(CH_2_) at 1270 cm^−1^ and 1446 cm^−1^ were found on all of the coatings. C–H vibration, CH_3_ bending and –O–CH_2_– deformation vibration were detected in 0.8TiO_2_/B72 and 1.0TiO_2_/B72 coatings (Figure 4a and Table 3). The peak of 1717 cm^−1^ was carbonyl group in α, β-unsaturated acids and detected in all coatings which corresponding well with the chemical composition of Paraloid B72. After UV irradiation for 1 day, many characteristic peaks disappeared with the loading of TiO_2_ especially at the high loading of TiO_2_ (Figure 4b and Table 3). Most of the peaks were found in Paraloid B72 and 0.2TiO_2_/B72 coating after 1-day UV aging while the peaks located at 1160 cm^−1^ and 1270 cm^−1^ were disappeared (Table 3). This result illustrates that Paraloid B72 was not decomposed after UV irradiation for 1 day and adding 0.2 wt% TiO_2_ did not trigger serious adverse effect on Paraloid B72 which might be attributed to the insufficient of generated free radicals from TiO_2_ under UV light. However, with the increment of TiO_2_ in Paraloid B72, the photo aging became more apparent which might be attributed to the generation of more free radicals leading to the degradation of Paraloid B72. Continually prolonging UV photo aging time, Paraloid B72 and TiO_2_/B72 coating with 0.5 wt% and 0.8 wt% TiO_2_ was degraded and only the carbonyl group in α, β-unsaturated acids can be found at 1717 cm^−1^. Moreover, 1.0TiO_2_/B72 was degraded completely after 7 days’ UV irradiation because there were not any characteristic peaks can be overserved in its FTIR spectrum even the 1717 cm^−1^ peak disappeared (Figure 4c and Table 3). Interestingly, some of the characteristic peaks were still found in 0.2TiO_2_/B72 coating indicating that adding 0.2 wt% TiO_2_ protected the Paraloid B72 from UV decomposition (Figure 4c and Table 3). The results corresponded well with water contact angle that after 7 days UV irradiation, water contact angle on 0.2TiO_2_/B72 coating is the largest and the water contact angle decrement was smaller compared with the water contact angle on other coating after 7 days’ irradiation (Figure 4d). The peak intensity at 1717 cm^−1^ of both B72 and 0.5 TiO_2_/B72 coatings decrease with the increasing of UV irradiation time. This result verified the effects of UV light and the utilization of TiO_2_ nanoparticles on the aging of Paraloid B72 (Appendix A).

### 3.5. SEM Characterization of TiO_2_/B72 Coatings under UV Irradiation

Since exposure under UV light leads to the degradation of Paraloid B72 and TiO_2_/B72, UV irradiation might cause the morphology change. To verify this, SEM was performed on Paraloid B72 coating and TiO_2_/B72 coatings after 5 days’ UV irradiation (Figure 5) showing that pores were blocked and some of the pores were disappeared at high loading of TiO_2_. In details, pores in Paraloid B72 transformed from open pores into half or closed pores (Figure 5a). This phenomenon is more significant in 0.2TiO_2_/B72 coatings after 5 days of UV irradiation that the pore became smaller and the number of pores decreased (Figure 5b) showing that the added TiO_2_ caused a certain degradation of Paraloid B72. However, after 5 days UV irradiation the pores were difficult to be found in 0.5TiO_2_/B72, 0.8TiO_2_/B72 and 1.0TiO_2_/B72 coatings (Figure 5c–e). This might be because more free radicals were generated from the higher content of TiO_2_ in Paraloid B72. The high activity and energy of free radicals e.g., OH•, superoxide radical can decompose some chemical bonds such as C-H, C-OH, C-O etc. Though adding certain amount of TiO_2_ into Paraloid B72 contributed to the degradation of organic pollutants, the generated extra free radicals from TiO_2_ under UV light irradiation may also decompose the polymer matrix leading to the photo aging of the polymer matrix and loss of protective functions.

Paraloid B72 was photo-degraded more seriously after UV light irradiation of 7 days (Figure 6). The pores vanished on all of the specimens and some papillae were generated on TiO_2_/B72 coatings after UV aging for 7 days (Figure 6b–e). Unlike the original 1.0TiO_2_/B72 coating whereby theTiO_2_ nanoparticles were buried into the Paraloid B72, TiO_2_ seems extracted from Paraloid B72 and dispersed on the surface (Figure 6e). This might due to the higher content of TiO_2_ in Paraloid B72 and generated more free radicals under UV light for longer time. As a consequence, the loading amount of TiO_2_ in Paraloid B72 should be well controlled, high loading of TiO_2_ lead to the degradation of Paraloid B72 which is not suitable for heritage protection coatings.

### 3.6. Optical Profilometer Analysis of TiO_2_/B72 Coatings after UV Irradiation

We selected 0.5TiO_2_/B72 as a model to illustrate the change of surface roughness before and after UV irradiation (Figure 7). As shown in Figure 7(a1), the surface of 0.5TiO_2_/B72 is relatively smooth and the roughness line in Figure 7(a2) reflects the fact that the surface is smooth without apparent peaks and a significant valley was detected. After irradiation for 5 days under UV light, the surface became rough with many particles dispersed on the surface (Figure 7(b1)) compared with the surface before UV irradiation (Figure 7(a1)). The roughness analysis indicates that there many small peaks formed on the surface after 5 days UV irradiation and the height ranges from nanometers to several micrometers (Figure 7(b2)). Continually increasing the UV irradiation time, many small particular-like structures were observed on 0.5TiO_2_/B72 coatings after 7 days’ UV irradiation (Figure 7(c1)). This was confirmed by the roughness line analysis that the surface is composed with many small peaks with the height less than 2 μm (Figure 7(c2)) and larger peaks can also be found which might be due to the TiO_2_ nanoparticles being extracted from the matrix after long UV irradiation (Figure 7(c2)).

To accurately evaluate the effects of irradiation time on the surface morphology and surface roughness, three-dimensional (3D) surface profiles, the whole surface area average roughness, the arithmetic mean surface roughness (Ra), root-mean-square roughness (Rq), maximum peak height (Rp) and maximum valley depth (Rv) of 0.5TiO_2_/B72 coating under the UV irradiation of 0 day, 5 days and 7 days were measured (Figure 8). As shown in Figure 8a, the surface of the original 0.5TiO_2_/B72 was relatively smooth with several small peaks which might be attributed to the aggregation of TiO_2_ nanoparticles. After being irradiated for 5 days and 7 days, the 3D surface profiles show that there many aggregations appeared and the aggregations became smaller, finer and higher with the increasing UV irradiation time (Figure 8b,c). The average roughness of 0.5TiO_2_/B72 coatings was measured showing that the average roughness decreased from 14 μm to 8 μm with the UV irradiation time increasing from 0 day to 7 days (Figure 8d). The reduction of average roughness might be because the longer UV irradiation leading to the generation of more free radicals which cause the degradation of Paraloid B72. To further evaluate the surface roughness, Ra and Rq increased with prolonging UV irradiation time (Figure 8e) showing that the Ra increased from 387. 5 nm to 731.5 nm while the Rq increased from 734.6 nm (0 day) to 1000.7 nm (7 days irradiation) indicating the surface is composed with more peaks. With the increment of UV irradiation, both the height of peaks and the depth of valleys decreased (Figure 8f) indicating that the longer irradiation caused the vanish of valleys and the peaks. This is ascribed to more free radicals generated from TiO_2_ nanoparticles under UV irradiation. These free radicals broke the structure of Paraloid B72 caused the degradation of Paraloid B72 and the reorganization of the polymer matrix, thereby fining the surface structure.

### 3.7. Water Adsorption of TiO_2_/B72 Coatings Before and after Photo Aging

The adverse effects of the TiO_2_ photocatalytic property on the polymer matrix can also be verified by characterizing the water adsorption performance of TiO_2_/B72 coatings after UV irradiation for different days. As shown in Figure 9, the water adsorption capacity of sandstone was 41 mg/(h·m^2^) and it decreased to 35 mg/(h·m^2^) after it was coated with Paraloid B72 and the water adsorption capacity continuously dropped with the addition of TiO_2_ nanoparticles and UV irradiation. This result illustrates that adding TiO_2_ nanoparticles into Paraloid B72 caused the decrease of water permeation capacity. This can be explained by the photocatalytic property of TiO_2_, in that the photo generated free radicals not only decomposed the deposited organic compounds at the surface but also degraded the polymer matrix leading to the reduction of pore numbers and the pore density on the coatings.

## 4. Conclusions

This work explored the effects of TiO_2_ nanoparticles on the pore structure of Paraloid B72 coating and the photo-aging behaviors of TiO_2_/Paraloid B72 under UV irradiation. Results showed that the size and number of the pores (which were generated in Paraloid B72) decreased with the addition of more of TiO_2_ nanoparticles. After incorporating more than 0.5 wt% TiO_2_ nanoparticles, the pores in Paraloid B72 vanished and the surface became rough with many significant TiO_2_ nanoparticle aggregations. Moreover, the incorporated TiO_2_ nanoparticles might generate adverse effects on Paraloid B72. Results showed that the added TiO_2_ nanoparticles caused the degradation of Paraloid B72 after exposure to UV light for a certain time and the pores were reduced by prolonging the UV irradiation time. The UV irradiation caused the increase of surface roughness and the TiO_2_ nanoparticles were extracted from the matrix after long UV irradiation. This degradation behavior of Paraloid B72 caused the reduction of water adsorption capacity which is not suitable for stone heritage relics’ protective coating.

## Figures and Tables

**Figure 1 polymers-13-00262-f001:**
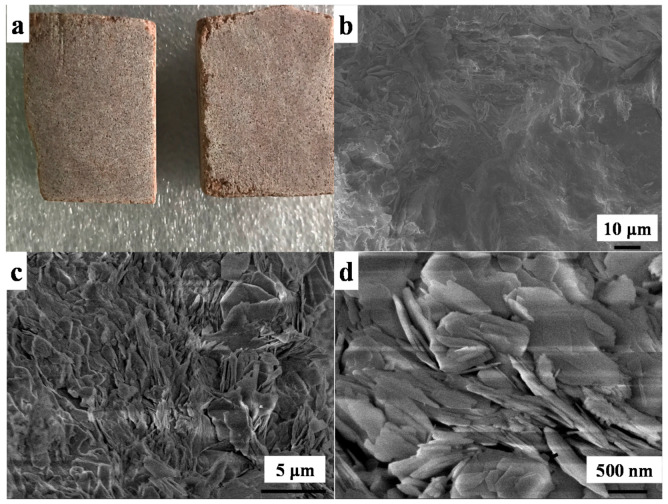
Digital and scanning electron microscope (SEM) images of the utilized sandstone in this work; (**a**): digital image of sandstone; (**b**–**d**) surface morphology of sandstone under different magnification.

**Figure 2 polymers-13-00262-f002:**
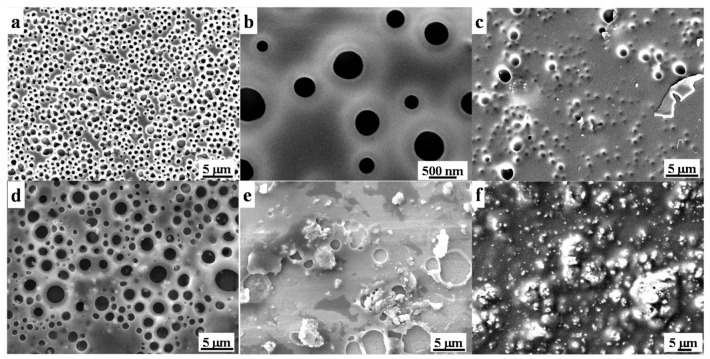
SEM images of these as-prepared coatings; (**a**,**b**): Paraloid B72; (**c**–**f**): TiO_2_/Paraloid B72 coatings with varied content of TiO_2_ corresponding to TiO_2_ nanoparticles content of 0.2%, 0.5%, 0.8% and 1.0%, respectively.

**Figure 3 polymers-13-00262-f003:**
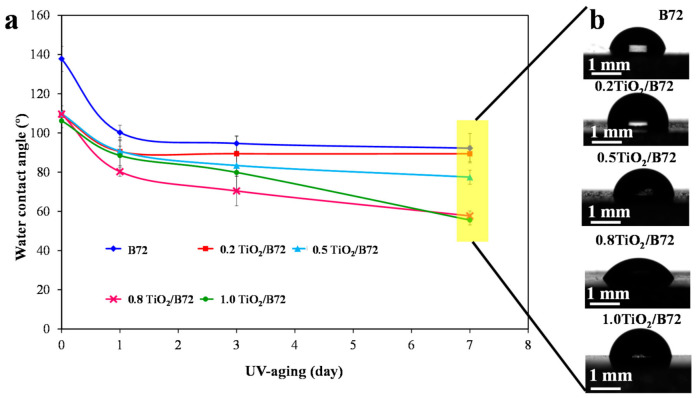
Surface wettability characterization via measuring the water contact angle; (**a**): the effects of TiO_2_ loading on the wettability of Paraloid B72 after UV light irradiation; (**b**): digital images of the water contact angles on TiO_2_/Paraloid B72 composite coatings.

**Figure 4 polymers-13-00262-f004:**
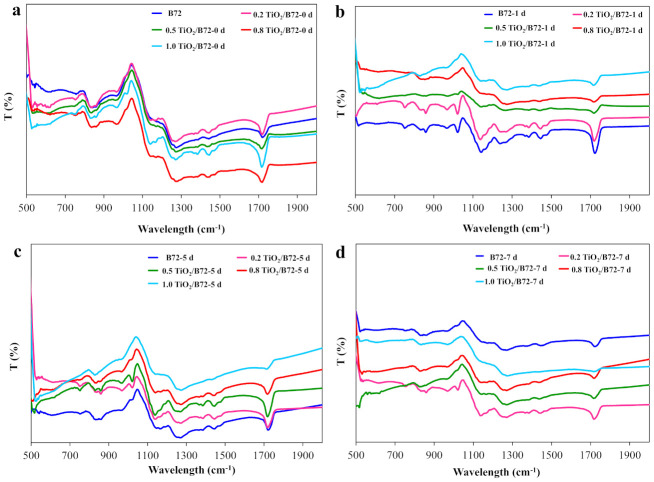
Fourier transform infrared (FTIR) spectroscopy characterization of TiO_2_/Paraloid B72 composite coating before and after UV irradiation; (**a**): 0 day; (**b**): 1 day UV irradiation; (**c**): 5 days UV irradiation; (**d**): 7 days UV irradiation.

**Figure 5 polymers-13-00262-f005:**
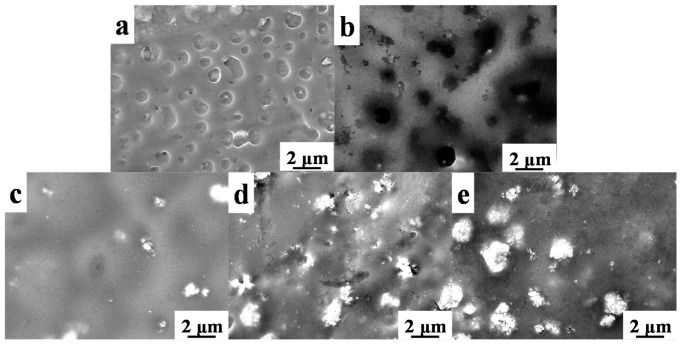
SEM images of TiO_2_/B72 after UV irradiation for 5 days; (**a**): B72 coating; (**b**): 0.2TiO_2_/B72 coating; (**c**): 0.5TiO_2_/B72 coating; (**d**): 0.8TiO_2_/B72 coating; (**e**): 1.0TiO_2_/B72 coating.

**Figure 6 polymers-13-00262-f006:**
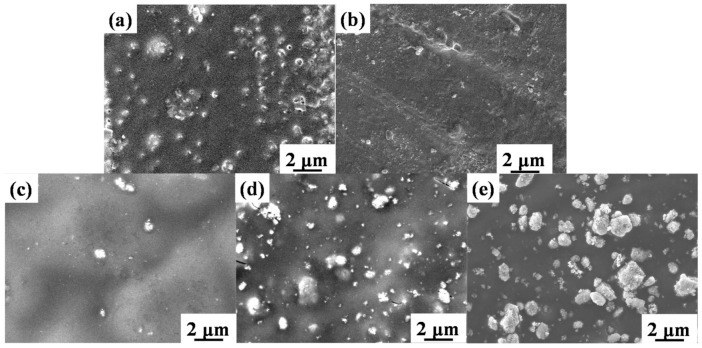
SEM images of TiO_2_/B72 coatings under UV irradiation for 7 days; (**a**): B72; (**b**): 0.2TiO_2_/B72; (**c**): 0.5TiO_2_/B72; (**d**): 0.8TiO_2_/B72; (**e**): 1.0TiO_2_/B72.

**Figure 7 polymers-13-00262-f007:**
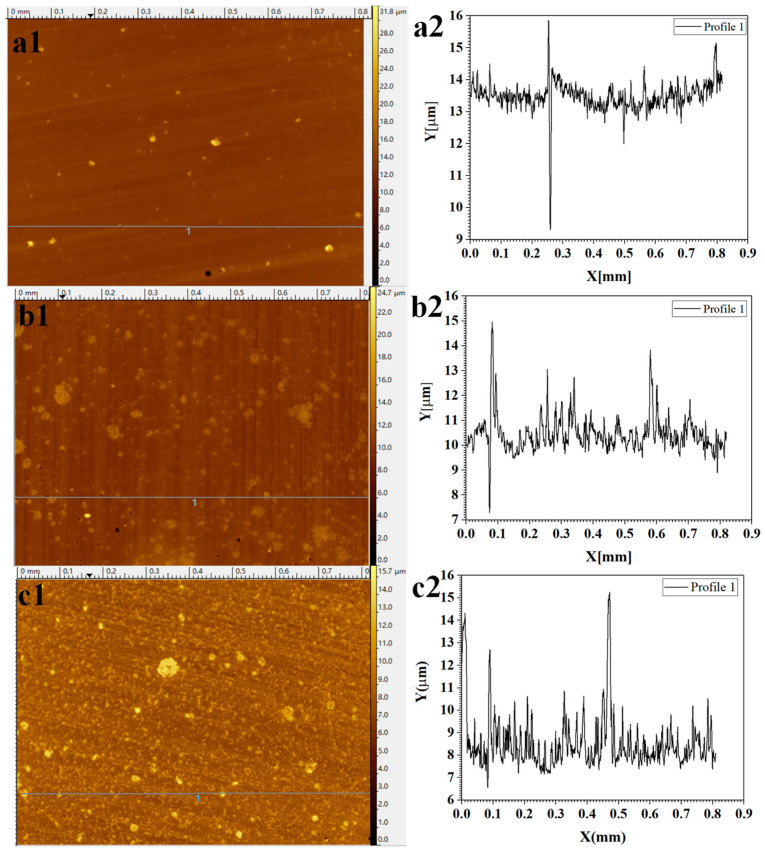
Optical profilometer analysis of 0.5TiO_2_/B72 coatings under varied UV irradiation time; (**a1**,**a2**): surface morphology and roughness analysis of 0.5TiO_2_/B72 coating before UV irradiation; (**b1**,**b2**): surface morphology and roughness analysis of 0.5TiO_2_/B72 coating after UV irradiation for 5 days; (**c1**,**c2**): surface morphology and roughness analysis of 0.5TiO_2_/B72 coating after UV irradiation for 7 days.

**Figure 8 polymers-13-00262-f008:**
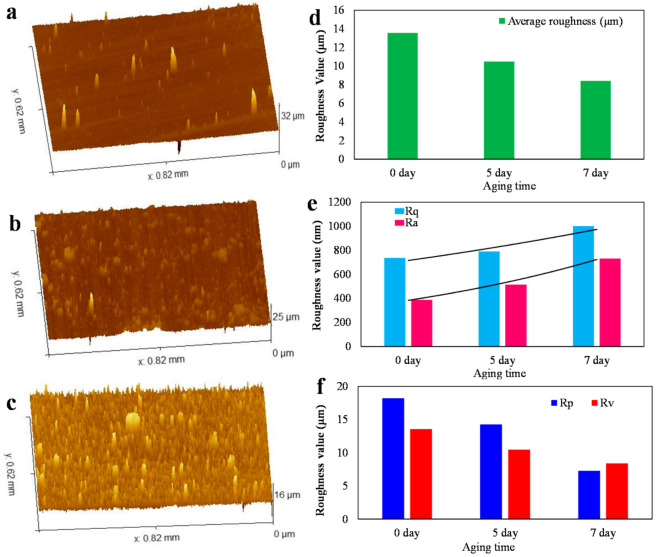
Surface structure and roughness analysis of 0.5TiO_2_/B72 coatings before and after UV irradiation; (**a**), (**b**) and (**c**): 3D surface morphology of 0.5TiO_2_/B72 coating irradiated under UV light for 0 day, 5 days and 7 days, respectively; (**d**) to (**f**): average roughness, arithmetic mean surface roughness (Ra), root-mean-square roughness (Rq), maximum peak height (Rp) and maximum valley depth (Rv) of 0.5TiO_2_/B72 coating under the UV irradiation of 0 day, 5 days and 7days, respectively.

**Figure 9 polymers-13-00262-f009:**
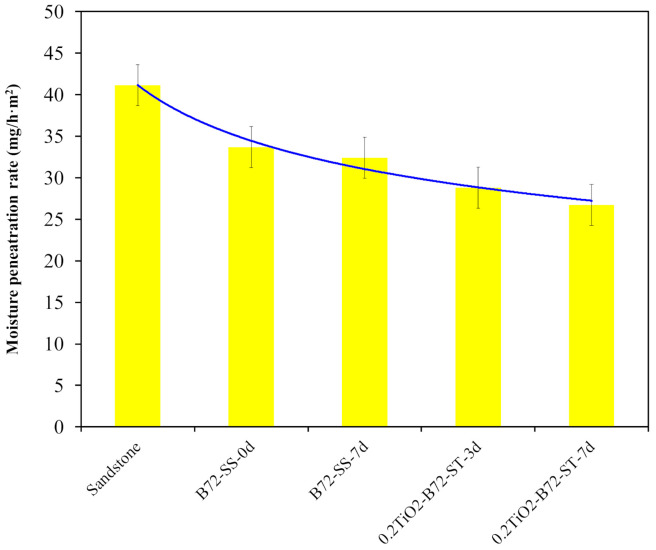
Water adsorption capacity of sandstone, sandstone coated Paraloid B72, and 0.2TiO_2_/B72 after UV irradiation.

**Table 1 polymers-13-00262-t001:** Sample information of TiO_2_/Paraloid B72 hybrid coating.

Sample ID	B72	0.2 TiO_2_/b72	0.5 TiO_2_/B72	0.8 TiO_2_/B72	1.0 TiO_2_-B72
P25 content (wt%)	0	0.2	0.5	0.8	1.0

**Table 2 polymers-13-00262-t002:** Chemical composition of the utilized sandstone.

Comp.	SiO_2_	Al_2_O_3_	Na_2_O	CaO	K_2_O	Fe_2_O_3_
wt%	84.20	1.78	1.26	0.44	0.41	0.20
Statistic error (%)	0.18	0.90	1.55	1.20	1.20	0.84

**Table 3 polymers-13-00262-t003:** FTIR analysis of TiO_2_/Paraloid B72 coatings before and after UV irradiation.

Wavelength (cm^−1^)	0 day					1 day					7 day				
	B72	0.2TiO_2_/B72	0.5TiO_2_/B72	0.8TiO_2_/B72	1.0TiO_2_/B72	B72	0.2TiO_2_/B72	0.5TiO_2_/B72	0.8TiO_2_/B72	1.0TiO_2_/B72	B72	0.2TiB	0.5TiO_2_/B72	0.8TiO_2_/B72	1.0TiO_2_/B72
758	√	√	√	√	√	√	√	-	-	-	-	√	-	-	-
836	√	√	√	√	√	√	√	-	-	-	-	-	-	-	-
858	√	√	√	√	√	√	√	-	-	-	-	√	-	-	-
968	√	√	√	√	√	√	√	-	-	-	-	√	-	-	-
1020	-	-	-	-	√	√	√	decrease	-	-	-	√	-	-	-
1140	-	√	√	√	√	√	√	-	-	-	-	√	-	-	-
1160	-	√	√	√	√	-	-	-	-	-	-	-	-	-	-
1270	√	√	√	√	√	-	-	-	-	-	-	-	-	-	-
1366	-	-	-	-	√	√	√	-	-	-	-	weak	-	-	-
1385	-	-	-	-	√	√	√	-	-	-	-	weak	-	-	-
1446	√	√	√	√	√	√	√	-	-	-	-	weak	-	-	-
1475	-	-	-	√	√	√	√	-	-	-	-	-	-	-	-
1717	√	√	√	√	√	√	√	√	√	√	√	√	√	√	disappear

## Data Availability

The data presented in this study are available on request from the corresponding author.

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
