# Peer review of "The Adverse Effects of TiO2 Photocatalycity on Paraloid B72 Hybrid Stone Relics Protective Coating Aging Behaviors under UV Irradiation"

_polymers, 2021, doi:10.3390/polym13020262_

Round 1
Reviewer 1 Report
The authors try to investigate the effect of TiO2 on the UV-aging of paraloid B72. The coated sand stones are investigated by SEM, contact angle measurement and IR-spectroscopy. The paper, especially the figures, but also the evaluation of some measurements are not carefully prepared.
Fig. 2 , the scaling is inconsistant, so a sensible comparison of the samples is not possible.
In Fig. 7 the scaling is much too small and not readable.
Fig. 4 the unit „wavelength (nm-1)“ is nonsense! Should be wavenumer (cm-1)! But also the interpretation of Fig. 4 makes no sense, because the peak at 1715 cm-1 is increasing (what‘s not possible!) for the samples B72 and 0.5/B72 ! Table3 a legend is missed.
There is really a big difference between Fig. 5a+6a, which is not believable!
Fig.9 the part „SS“ and „ST“ in sample name is not explained!
The order oft he samples in Fig.3 is not allegeable. Why should the samples with 1% TiO2 and 0.2% TiO2 should show a similar behavior, but the samples without TiO2 and with 0.8% TiO2 a different behavior? And in Fig.3 the experimental error is missed, which shouldn’t be so small for contact angle measurements.
By the way, it is well known, that the contact angle of anatase (P25) is decreasing during UV-radiation! So this correlation should be also considered and discussed!
Experimental: The sample preparation ist not clear: the amount of 0.2, 0.5, 0.8, 1.0wt% TiO2, is it the amount, added to the coating solution or the amount in the resulting material? So is the amount of TiO2 related to the solution or is it related to B72? Looking to Fig.2f, it seems, that there is much more than only 1% TiO2 in the coating.
The IR-spectroscopy, how is it performed? Is the reflection of the coating measured?
There is now consistant message in the paper and I miss a statement to the pores, are they desired for a kind of breathing, that the samples can dry or is the aim to avoid pores?
Reviewer 2 Report
This is an interesting and original article on TiO2/Paraloid B72 hybrid coating under UV light irradiation. The authors very clearly determined the aim of the study. The paper is well written - clear motivation, explanations, references are correct. The procedures and methodology are described in details and are clear.
However, before allowing for publication, I would suggest making some minor improvements:
- What was the temperature during UV aging?
- line 100 – “3 to 5 samples were prepared” for which sample ID were 3 samples and for which 5?
- line 199 – please add the superscript for the area unit.
- Figure 2 – if possible, please standardize the magnification in the photos, then it is easier to compare them.
- Figure 3 – It is worth arranging figures "b" according to the sequence of graphs in figure "a" for 7 days, looking from the top: 0,2TiO2/B72, 0,5TiO2/B72, 1,0TiO2/B72, B72, 0,8TiO2/B72.
- Figure 6 – one photo has a different magnification, if possible, please make it uniform.
- Figure 7 – Please increase the descriptions of axes and scale for drawings a2, b2, c2. In addition, the description under the drawing can be shortened to make it more readable, e.g. "Optical profilometer analysis of 0.5TiO2/B72 coatings: a) before UV irradiation, b) for 5 days, c) for 7 days, (a1, b1, c1 - surface morphology; a2, b2, c2 - roughness analysis) ". Then "a", "b", "c" should be added above each pair of pictures.
- line 277 “wea” ?
After completing the above mentioned comments, I strongly recommend the paper for publication.
Reviewer 3 Report
The authors report on the effect of TiO2 loading in photocatalytic titania-polymer composite protective coatings for sandstone relics. Different amounts of commercial P25 nanopowder were mixed in Paraloid B72 resin and applied as sandstone coatings under UVC irradiation. The composite coatings were investigated by several methods including SEM, contact angle measurements, FTIR and water adsorption as a function of TiO2 content and duration of UV exposure. Overall, the work is well aimed reporting interesting results on the adverse photocatalytic effects of titania inclusions in polymer coatings. However, the following points should be considered:
What is the power density of UVC irradiation? How do these irradiations compare with real weathering conditions under solar light? These points should be reported and discussed as the degradation effects may be partly due to the high energy of UVC light.
For some Figures (e.g. 2, 4 and 5) it is not clear if the presented results concern the coated stones or bare coatings.
Is the photocatalytic activity of the optimum 0.2 TiO2/b72 composite coating with low P25 content, adequate for organic pollutant degradation?
Is it possible to improve titania dispersion in the polymer matrix and reduce aggregation effects as the P25 loading increases?
